# Autoimmune Mast Cell Activation Test as a Diagnostic Tool in Chronic Spontaneous Urticaria

**DOI:** 10.3390/ijms25179281

**Published:** 2024-08-27

**Authors:** Ana Koren, Luka Dejanović, Matija Rijavec, Peter Kopač, Mojca Bizjak, Mihaela Zidarn, Mitja Košnik, Peter Korošec

**Affiliations:** 1University Clinic of Respiratory and Allergic Diseases Golnik, 4204 Golnik, Slovenia; luka.dejanovic@klinika-golnik.si (L.D.); matija.rijavec@klinika-golnik.si (M.R.); peter.kopac@klinika-golnik.si (P.K.); mojca.bizjak@klinika-golnik.si (M.B.); mihaela.zidarn@klinika-golnik.si (M.Z.); mitja.kosnik@klinika-golnik.si (M.K.); peter.korosec@klinika-golnik.si (P.K.); 2Biotechnical Faculty, University of Ljubljana, 1000 Ljubljana, Slovenia; 3Faculty of Medicine, University of Ljubljana, 1000 Ljubljana, Slovenia; 4Faculty of Pharmacy, University of Ljubljana, 1000 Ljubljana, Slovenia

**Keywords:** mast cell activation test, LAD2, chronic spontaneous urticaria, omalizumab, CD63 expression

## Abstract

Chronic spontaneous urticaria (CSU) is associated with skin mast cell activation, and its triggering mechanisms are not completely elucidated. Evidence suggests an autoimmune component of CSU. Our aim was to assess the usefulness of an autoimmune mast cell activation test (aiMAT) for diagnosing and differentiating CSU into different subtypes. We enrolled 43 patients with active, uncontrolled CSU before starting treatment with omalizumab and 15 controls. Patients were evaluated based on omalizumab response. aiMATs were performed using non-IgE-sensitized (NS) or myeloma IgE-sensitized (S) LAD2 cells, which were then stimulated with CSU/control sera (25 µL and 10 µL). The expression of CD63 was assessed with flow cytometry. CD63 response on NS-LAD2 was significantly increased in CSU patients compared to controls after the stimulation with 25 µL CSU/control sera (*p* = 0.0007) and with 10 µL CSU/control sera (*p* = 0.0001). The ROC curve analysis demonstrated an area under the curve (AUC) of 0.82. The cutoff for autoimmune-non-IgE-sensitized-MAT was 40.3% CD63+ LAD2, which resulted in 73.3% sensitivity and 81.4% specificity. CD63 response on S-LAD2 was significantly increased in CSU patients compared to controls after the stimulation with 25 µL CSU/control sera (*p* = 0.03). The ROC curve analysis demonstrated an AUC of 0.66. The cutoff for the autoimmune-myeloma IgE-sensitized-MAT was 58.4% CD63+ cells, which resulted in 62.8% sensitivity and 66.7% specificity. Overall, 36 out of 43 (84%) patients responded to omalizumab, and 7 (16%) were nonresponders. We found no differences between LAD2 CD63 response and response to omalizumab. In conclusion, aiMAT could represent a new diagnostic tool in CSU. Additional studies are needed to evaluate the potential benefits during omalizumab therapy.

## 1. Introduction

Chronic spontaneous urticaria (CSU) is a skin condition characterized by recurrence of wheals and/or angioedema for more than 6 weeks [1]. Mast cell (MC) activation and degranulation with subsequent release of mediators of inflammation plays a significant role in CSU [2]. Still, the exact cause of MC activation in CSU is unclear [3]. The pathogenesis of CSU is increasingly recognized to be driven by two main autoimmune mechanisms that can be characterized by the presence of IgE autoantibodies to autoallergens (type I) or IgG autoantibodies targeting either IgE or the high-affinity IgE receptor (FcɛRI) on MCs and basophils (type IIb) [3]. These autoantibodies are thought to activate MCs and basophils via the IgE/FcɛRI pathway, releasing histamine and other inflammatory mediators, which in turn cause the characteristic signs and symptoms of CSU [4].

Treatment of CSU comprises a stepwise approach that primarily involves second-generation H1-antihistamines (sgAHs) following dosage escalation in case of poor response. Omalizumab is a recombinant humanized monoclonal antibody that targets IgE. It is approved for treating CSU in patients who have not responded adequately to sgAHs [1,5]. Omalizumab binds to the free IgE, reducing its availability to binding to FcɛRI and downregulating FcɛRI on MCs and basophils [6]. However, not all CSU patients respond to omalizumab equally; some respond in the first week after the initial application, while others respond after several weeks [7]. A subgroup of patients does not respond at all and is characterized by very low basophil counts, positive basophil histamine release assay (BHRA), and positive basophil activation test (BAT) with donor basophils [8,9,10].

The mast cell activation test (MAT) is a cellular test used to evaluate the functional status of IgE antibodies by measuring the responsiveness of human MCs (primary blood-derived or cell line) to various stimuli [11]. Recent studies have shown that MAT could be used to diagnose food, drug, and insect venom allergies [11,12,13,14], monitor oral immunotherapy [15], and diagnose patients with uninterpretable BAT caused by nonresponding basophils [14]. The concept of autoimmune MAT (aiMAT) has been proposed [16], but no studies have evaluated MAT’s usefulness in CSU so far. LAD2 is a human MC line [17]. The expression of FcԑRI on LAD2 cells and the ability to bind IgEs after passive sensitization suggests that LAD2 can serve as a relevant cell model system for evaluation of the autoimmune activity of anti-FcRI and anti-IgE autoantibodies in CSU. 

Our study aimed to assess the usefulness of aiMAT on non-IgE-sensitized or myeloma IgE-sensitized LAD2 cells (NS-LAD2 or S-LAD2, respectively) to evaluate aiMAT as a diagnostic tool in CSU and to differentiate its subtypes based on the response to omalizumab treatment.

## 2. Results

### 2.1. Patient and Treatment Characteristics

The median age of patients was 48 years (range, 22–81) years, and 30 out of 43 (70%) were women. The median 7-day Urticaria Activity Score (UAS7) before the start of omalizumab treatment was 28 (range, 12–42). The median age of healthy controls was 45 years (range 27–74) years; 11/15 (73%) were women. In total, 36 (84%) and 7 (16%) out of 43 patients were responders (OMA-R) and nonresponders (OMA-NR) to omalizumab, respectively. Among the former, 23 (53%) were classified as early complete responders and 13 (30%) as late complete responders. The demographic, clinical, and laboratory features of patients and controls are summarized in Table 1.

### 2.2. LAD2 Characteristics

For quality control assessment, LAD2 cells were analysed for CD117, FcɛRI, and IgE expression in each of the four experimental batches. LAD2 cells expressed CD117 and FcԑRI on their surface. After the sensitization with human myeloma, IgE, IgE+ LAD2 were detected in high proportions (median 93.7%, range 86.0–95.2), whereas no IgEs were detected on NS LAD2 cells (median 0.7%; range 0.0–0.94). After the stimulation with CSU patient sera, no differences in IgE expression on the LAD2 cell surface were observed. The representative pictures of flow cytometric analysis of CD117, FcɛRI, and IgE expression on NS-LAD2 or S-LAD2 are shown in Figure 1.

### 2.3. The Diagnostic Utility of Autoimmune LAD2 MAT

The median level of response to positive control anti-IgE on S-LAD2 was 47.8% (range 40.0–63.2)), and the response to medium alone was 3.5% (range 2.5–5.3). The median level of response to positive control anti-IgE on NS-LAD2 was 1.5% (range 1.5–3.3), and the response to medium alone was 2.3% (range 2.2–2.8). The results of the diagnostic utility of aiMAT are shown in Figure 2. Using NS-LAD2 cells (left side of the panel), there were significant differences between CSU patients and healthy controls after the stimulation with 25 µL of sera (median CD63 response 64.9% vs. 54.2%; *p* = 0.0007) and after the stimulation with 10 µL of sera (median CD63 response 52.5% vs. 35.3%; *p* = 0.0001). The ROC curve analysis demonstrated an area under the curve (AUC) of 0.78 (95% CI 0.64–0.92) after the stimulation with 25 µL of sera and AUC of 0.82 (95% CI 0.69–0.95) after the stimulation with 10 µL of sera. Using S-LAD2 cells (right side of the panel), there were significant differences between CSU patients and healthy controls after the stimulation with 25 µL of sera (median CD63 response 62.5% vs. 52.6%; *p* = 0.03), whereas no differences were found after the stimulation with 10 µL of sera (median CD63 response 36.1% vs. 30.3%; *p* = 0.1). The ROC curve analysis demonstrated an AUC of 0.66 (95% CI 0.48–0.83) after the stimulation with 25 µL of sera. Spearman’s coefficient correlation analysis between aiMAT results after the stimulation with 25 µL of sera and after the stimulation with 10 µL of sera showed significant correlations using both NS-LAD2 (R_s_ = 0.48; *p* = 0.001) and S-LAD2 (R_s_ = 0.78; *p* < 0.0001) cells (Figure 3).

The optimal cutoff for the ai-NS-MAT (autoimmune non-IgE sensitized MAT) after the stimulation with 10 µL of sera was 40.3% CD63+ LAD2 cells, which resulted in 73.3% sensitivity and 81.4% specificity in identifying CSU patients. The optimal cutoff for the ai-S-MAT (autoimmune myeloma IgE sensitized MAT) after the stimulation with 25 µL of sera was 58.4% CD63+ LAD2 cells, which resulted in 62.8% sensitivity and 66.7% specificity in identifying CSU patients. For detailed information on diagnostic performance, refer to Table 2.

Overall, 35 out of 43 CSU patients (81%) tested positive for ai-NS-MAT, and 27 out of 43 (63%) tested positive for ai-S-LAD2 MAT. Moreover, 26 out of 43 (60%) were positive for both ai-NS-MAT and ai-S-MAT, whereas 36/43 (84%) tested positive for either test. ai-NS-MAT-positive patients, compared to ai-NS-MAT-negative ones, had significantly lower female gender distribution (3/8 [38%] vs. 27/35 [77%]; *p* = 0.03), whereas no significant differences were observed in other characteristics. ai-S-MAT-positive patients had significantly higher age (53 vs. 43 years; *p* = 0.002), baseline UAS7 (32 vs. 21; *p* = 0.04), and lower total serum IgE levels (42 vs. 126 kU/L; *p* = 0.008) and absolute basophil count (3 vs. 12 cells/µL; *p* = 0.02) compared to ai-S-MAT-negative patients. No significant differences were observed in gender distribution, body mass index (BMI), disease duration, absolute numbers of blood lymphocytes, monocytes and granulocytes, and donor basophil activation (Table 3).

### 2.4. Predictive Value of LAD2 MAT for Omalizumab Therapy Response

The comparison of aiMAT results between CSU patients who responded to omalizumab treatment (early complete responders and late complete responders; OMA-R) and CSU patients who did not respond to omalizumab (nonresponders; OMA-NR) showed that there were no statistical differences between these two groups. However, in the case of ai-NS-MAT, we found significant differences between OMA-R and controls after the stimulation with 25 µL of sera (median CD63 response 66.0% vs. 54.2%; *p* = 0.0007) and 10 µL of sera (median CD63 response 51.9% vs. 35.3%; *p* = 0.0003). Also, a significant difference between OMA-NR and controls after the stimulation with 10 µL of sera (median CD63 response 55.7% vs 35.3%; *p* = 0.0003) was found (Figure 4).

## 3. Discussion

CSU is characterized by recurrent hives and angioedema, a result of inflammatory responses due to skin MC activation [1]. While the exact triggering mechanisms remain unknown, evidence suggests an autoimmune component [3]. Our results show that aiMAT, using NS-LAD2 or S-LAD2 stimulated with the CSU/control sera, distinguishes between CSU patients and controls and thus could represent a new diagnostic tool in CSU. Our results also show that aiMAT is not predictive of omalizumab treatment response. 

The current gold standard for autoimmune (type IIb) CSU diagnostics is a positive autologous serum skin test result in combination with functional bioassay (BHRA or BAT) and a positive immunoassay for specific IgG against FcɛRI and IgE [18]. While immunoassays are not usually available for routine clinical practice, basophil tests are considered the best in vitro tests for diagnosing autoimmune CSU [3]. However, the wider use of BAT is limited by several factors. These include the need for fresh healthy donor peripheral blood, which must be used within 24 h, significant variability among donors due to differences in FcɛRI expression [19], and the fact that about 10% of healthy individuals’ basophils are nonresponders to FcɛRI-dependent stimulation [19,20], thus not inappropriate as basophil donors. MAT could serve as a suitable alternative to BAT. The LAD2 MC line presents a promising cell source for MAT because it can be routinely maintained in the laboratory, does not have inter-donor variability, and expresses functional FcɛRI. Moreover, LAD2 cells can be transferred from one laboratory to another, thus enabling inter-laboratory comparisons. This makes LAD2 cells a consistent and reliable functional model for MC activation studies [17].

Functionally active autoantibodies to IgE and FcɛRI on MCs were first detected in the sera of CSU patients. Their functionality was defined by basophil and MC histamine release assays and BAT [21,22,23]. To recreate this condition, we used NS-LAD2 and S-LAD2 cells, which we stimulated with sera from CSU patients and controls. We assumed that NS-LAD2 would be a suitable model for the detection of anti-FcɛRI autoantibodies, while S-LAD2 would better detect anti-IgE autoantibodies. LAD2 quality control assessment confirmed that IgE from CSU patient/control sera did not bind to LAD2 cells during stimulation, suggesting CSU patients` IgEs are not responsible for LAD2 MC activation. This is in line with the results of previous studies which show that IgG autoantibodies targeting IgE and the FcɛRI in CSU are believed to activate tissue MCs [3]. To the best of our knowledge, this is the first study evaluating the diagnostic utility of MAT in CSU. Our results showed an increased LAD2 CD63 response after the stimulation with CSU sera, which was more strongly expressed in the case of stimulation of NS-LAD2 cells, which would potentially suggest a higher anti-FcɛRI than an anti-IgE autoimmune component in our CSU patient cohort. Overall, we found that 60% of patients were positive with both ai-NS-MAT and ai-S-MAT, whereas more than 80% of patients tested positive for ai-NS-MAT or ai-S-MAT. This result is in line with previous studies, which reported IgG-anti-FcɛRI and/or IgG-anti-IgE in up to 69% of CSU patients, with most studies reporting rates between 20% and 50% [4]. Our results support the hypothesis that CSU sera harbor autoimmune factors; this could be due to the presence of autoantibodies against IgE/FcɛRI or also against other MC activating factors, which are smaller than antibodies and are not associated with the FcɛRI/IgE pathway [24,25], potentially cell-activating chemokines. Other autoantibodies directed against MC-activating receptors, such as C5a or MRGPRX2, may also be involved. The degree to which these and other non-FcRI-mediated mechanisms contribute to mast cell degranulation in CSU has not yet been conclusively determined [26].

BAT using CD63 as a degranulation marker, performed from donor peripheral blood, has been proposed as a reliable functional assay in diagnosing CSU, which correlates with histamine release and autologous serum skin tests [27,28,29,30]. In our study, however, we found no differences in BAT with donor basophils results between patients who tested positive for ai-NS-MAT/ai-S-MAT and those who tested negative. Of note, a relatively high proportion of activated LAD2 in the control group indicates the presence of other serum factors that activate LAD2 cells and are not linked to CSU. This could be partially explained by a concept of conditional immunity involving natural antibodies against FcɛRI [31]. The presence of IgG–anti-FcεRI/IgE has also been previously found in other studies, representing up to 57% in healthy controls [32,33,34].

No differences in ai-NS-MAT or ai-S-MAT were observed between CSU patients who responded and patients who did not respond to omalizumab treatment, suggesting that aiMAT is not a predictor of omalizumab response. Our previous study showed that a low absolute basophil count is the strongest predictor of omalizumab nonresponders, followed by FcɛRI and IgE densities per basophils [8]. Omalizumab binds free IgE and consequently downregulates the expression of FcɛRI on MCs and basophils. The possible explanation is that while omalizumab targets the IgE/FcɛRI pathway, it does not target IgG autoantibodies to FcɛRI and other MC activating factors in CSU.

We are aware our study has several limitations. We used the LAD2 cell line as an MC source; an alternative could be primary human MCs obtained with differentiation from CD34+/CD117+ progenitors in the peripheral blood. Though utilizing cell lines possesses advantages like reproducibility and donor independence, it is known that primary human MCs are more susceptible to IgE-mediated degranulation than LAD2 cells [11]. A possible reason for some inconclusive results in our study might also be the relatively small sample size, with only seven OMA-NR. It might be that a larger patient group would provide more insightful data on how LAD2 MC activation and omalizumab treatment are related. Lastly, analysis of other MC mediators, such as β-hexosaminidase and several chemokines and cytokines, would provide us with additional information on how CSU sera impact MC activation.

To our knowledge, this is the first study evaluating the usefulness of aiMAT for diagnosing CSU and its differentiation into different subtypes based on omalizumab treatment response. Our findings support the role of MC activation in the pathogenesis of CSU and highlight the potential utility of aiMAT as a diagnostic tool in this condition. Further research is warranted to elucidate the association between MC activation and response to omalizumab and to further explore the utility of aiMAT in identifying predictors of therapeutic outcomes in CSU patients.

## 4. Materials and Methods

### 4.1. Patients and Treatment Evaluation

This study was performed from serum samples of 43 patients with severe CSU before starting treatment with omalizumab [8] and from 15 controls. Patients and controls were matched according to age, gender, and total IgE levels. Except for two patients who received three subcutaneous doses of 150 mg omalizumab, all patients received three subcutaneous doses of 300 mg omalizumab at four-week intervals. Response to treatment was defined as the UAS7 of ≤6 within two weeks (early complete responders) or 12 weeks (late complete responders), while no response was defined as UAS7 persisting at ≥7 within 12 weeks. All specimens were collected at the University Clinic of Respiratory and Allergic Diseases, Golnik, Slovenia. The MATs were performed from serum samples that were stored at −20 °C. The study was approved by the Slovenian Medical Ethics Committee (KME78/09/14), and all patients gave their informed consent.

### 4.2. LAD2 Cells

LAD2 cells were kindly donated from the National Institutes of Health, USA (National Institute of Allergy and Infectious Disease–Dr A.S. Kirshenbaum). Cells were cultured in StemPro-34 complete medium (CM), consisting of StemPro-34 SFM (Gibco–Thermo Fisher Scientific, Waltham, MA, USA) supplemented with 100 U/mL penicillin/streptomycin (Sigma-Aldrich, St Louis, MO, USA), 2 mM L-glutamine (Sigma-Aldrich, St Louis, MO, USA) and 100 ng/mL recombinant human stem cells factor (SCF, Stem Cell Technologies, Vancouver, Canada). Cells were cultured in a humidified incubator containing 5% CO_2_ at 37 °C, and the media was changed weekly.

### 4.3. Autoimmune LAD2 Mast Cell Activation Test

MAT was performed with LAD2 cells, which were either non-sensitized (NS) or sensitized (S) with 1 µg/mL human myeloma IgE (Millipore, Burlington, MA, USA) overnight. After washing, 20,000 cells were stimulated with 25 µL and 10 µL of CSU/control sera in the final volume of 150 µL at 37 °C for 15 min. The CSU/control serum volumes for stimulation were selected in a range that elicited dose-dependent LAD2 CD63 response. For the controls, the cells were exposed to medium alone (negative control) or 10 µg/mL anti-IgE mAb (positive control) (Sigma-Aldrich, St Louis, MO, USA). Degranulation was stopped by chilling on ice, after which APC anti-human CD63 (BioLegend, San Diego, CA, USA) was added and incubated for 20 min. Afterward, cell probes were washed twice and fixed before flow cytometry analysis using the FACSCanto II flow cytometer (BD Biosciences, Franklin Lakes, NJ, USA).

Sensitization of LAD2 was assessed with anti-IgE staining, and LAD2 quality was assessed by analyzing CD117 and FcɛRI expression in every batch of experiments. In case of the reduction of observed markers, a new batch of cells was thawed. LAD2 cells were stained with CD117-APC/Cyanine7 (BioLegend, San Diego, CA, USA), FcƐRIα-PE (BioLegend, San Diego, CA, USA), and IgE-FITC (Miltenyi Biotec, Bergisch Gladbach, Germany) or appropriate isotype staining controls (IgG1-APC/Cy7, IgG2b-PE, IgG1 (all from BioLegend, San Diego, CA, USA).

### 4.4. Total IgE Measurements

The total IgE and specific IgE against HBV and YJV from sera samples of patients were determined with the Immulite system (Siemens, München, Germany).

### 4.5. Enumeration of Basophils, Lymphocytes, Monocytes and Granulocytes

Enumeration of circulating basophils (CD123+HLA-DR− cells), monocytes, lymphocytes, and granulocytes in fresh whole blood was performed via flow cytometry as previously described [8,35]. 

### 4.6. BAT with Donor Basophils 

BAT with donor basophils was performed as previously described [8]. Briefly, 50 µL of healthy donor fresh whole blood was stimulated with 50 µL of CSU patient sera and CD63 expression on the surface of basophils was analyzed with flow cytometry.

### 4.7. Data Analysis

The data distribution was determined using the D’Agostino and Pearson omnibus tests. For comparisons between groups, we used the Mann–Whitney U test or unpaired t-test, as appropriate. The chi-square test was used to compare categorical values. Optimal cutoffs were determined using the Youden index. Flow cytometric analyses were performed using BD FACSDiva (version 8.0.1) (BD Biosciences, Franklin Lakes, NJ, USA) or FlowJo (version 10.7.2) (BD Biosciences, Franklin Lakes, NJ, USA) analysis software. Statistical analyses were conducted using GraphPad Prism (version 10.1.1) (GraphPad Software, Boston, MA, USA). A *p*-value below 0.05 was considered statistically significant. All reported *p*-values are two-tailed.

## Figures and Tables

**Figure 1 ijms-25-09281-f001:**
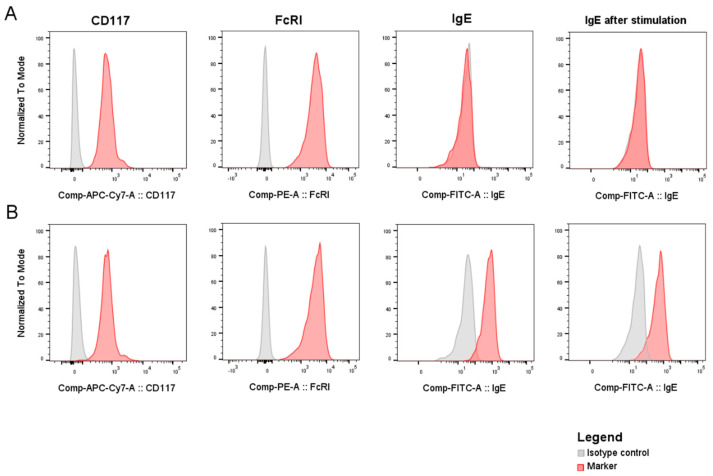
LAD2 immunophenotyping. Expression of CD117, FcɛRI, IgE, and IgE after stimulation with patients’/control sera on (**A**) non-IgE sensitized or (**B**) myeloma IgE sensitized LAD2 cells. APC-Cy7, Allophycocyanin/Cyanine7; PE, phycoerythrin; FITC, fluorescein isothiocyanate.

**Figure 2 ijms-25-09281-f002:**
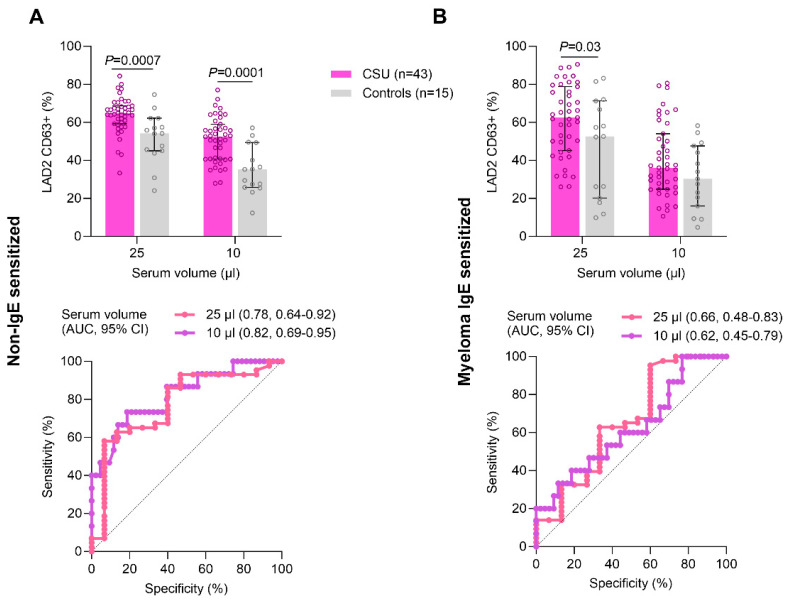
The diagnostic utility of autoimmune MAT in CSU. (**A**) Comparison in LAD2 CD63 response between 43 CSU patients and 15 controls after the stimulation of non-IgE sensitized LAD2 cells with 25 µL and 10 µL of sera and the corresponding receiver operator characteristic (ROC) curve. (**B**) Comparison in LAD2 CD63 response between CSU patients and controls after the stimulation of myeloma IgE sensitized LAD2 cells with 25 µL and 10 µL of sera and the corresponding ROC curve. The presented values are the median ± interquartile range.

**Figure 3 ijms-25-09281-f003:**
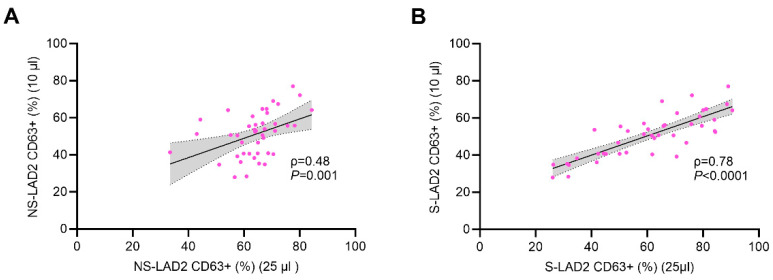
Spearman’s coefficient correlation analysis between LAD2 CD63 response after the stimulation with 25 µL CSU sera and the stimulation with 10 µL CSU sera using (**A**) non-IgE sensitized LAD2 cells (NS-LAD2) or (**B**) myeloma IgE sensitized LAD2 cells (S-LAD2) in 43 CSU patients.

**Figure 4 ijms-25-09281-f004:**
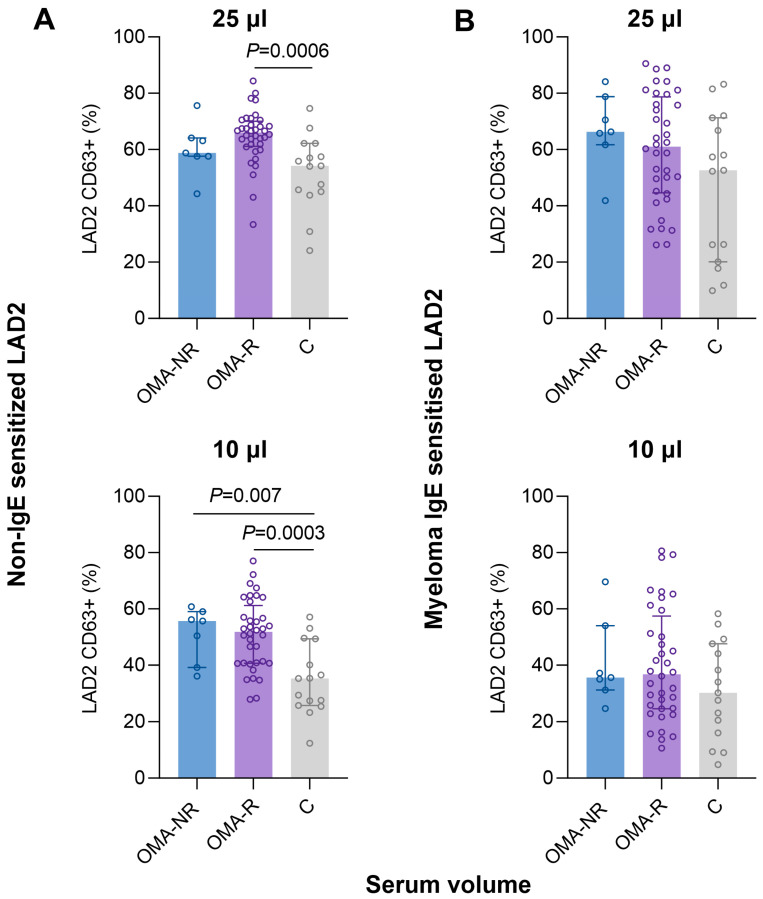
Differences in LAD2 CD63 response between nonresponders (OMA-NR) and responders (OMA-R) to omalizumab therapy and controls (C) after the stimulation of (**A**) non-IgE sensitized LAD2 or (**B**) myeloma IgE sensitized LAD2 cells with 25 µL (upper row) and 10 µL (lower row) of sera. The presented values are the median ± interquartile range.

**Table 1 ijms-25-09281-t001:** Demographic, clinical and laboratory characteristics of chronic spontaneous urticaria patients and controls.

Characteristics	CSU Patients	Controls	*p* Value
**Patients:** N	43	15	
**Age in years**: Median (range)	48 (22–81)	45 (27–74)	0.68
**Gender**: N (%)			
Male	13 (30.2)	4 (26.7)	0.79
Female	30 (69.8)	11 (73.3)	
**UAS7**: Median (range)	28 (12–42)	NA	
**Total IgE (kU/L**): Median (range)	60.5 (<1–1186)	51.3 (4–622)	0.99
**Disease duration in years**: Median (range)	3 (0–25)	NA	
**Response to omalizumab**: N (%)			
Early complete responders	23 (53.5)	NA	
Late complete responders	13 (30.2)	NA	
Nonresponders	7 (16.3)	NA	

CSU: chronic spontaneous urticaria; UAS7: 7-Day Urticaria Activity Score; NA: not applicable.

**Table 2 ijms-25-09281-t002:** Optimal cutoffs to classify subjects with chronic spontaneous urticaria.

	CSU Patients Versus Controls					
Parameters	Serum Volume (µL)	Cutoff	Sensitivity	Specificity	PPV	NPV
ai-NS-MAT	10	40.3	73.3 (44.9–92.1)	81.4 (66.6–91.6)	89.7 (78.9–95.4)	57.9 (40.7–73.4)
ai-S-MAT	25	58.4	62.8 (46.7–77.0)	66.7 (38.4–88.2)	84.4 (71.8–92.0)	38.5 (26.9–51.5)

*Note:* Optimal cutoffs were determined based on the Youden index. Sensitivity, specificity, PPV, and NPV with 95% CIs are indicated for each cutoff. CSU: chronic spontaneous urticaria; PPV, positive predictive value; NPV, negative predictive value; ai-NS-MAT: autoimmune non-IgE sensitized MAT; ai-S-MAT: autoimmune myeloma IgE sensitized MAT.

**Table 3 ijms-25-09281-t003:** Characteristics of chronic spontaneous urticaria patients, according to the autoimmune mast cell activation test results.

Characteristics	ai-NS-MAT (10 µL Sera)		*p* Value	ai-S-MAT (25 µL Sera)		*p* Value
	Positive	Negative		Positive	Negative	
Patients: N (%)	35 (81.4)	8 (18.6)		27 (62.8)	16 (37.2)	
Age in years:	49 (22–81)	40 (24–63)	0.13	53 (22–81)	43 (22–63)	**0.002**
Gender: N (%)						
Male	8 (18.6)	5 (11.6)	**0.03**	6 (13.9)	7 (16.3)	0.14
Female	27 (62.8)	3 (7.0)		21 (48.9)	9 (20.9)	
BMI (kg/m^2^)	26.9 (19.5–45.0)	27.8 (20.7–33.1)	0.94	26.9 (19.5–45.0)	26.5 (20.3–37.1)	0.58
UAS7	28 (12–42)	18 (14–42)	0.93	32 (17–42)	21 (12–42)	**0.04**
Disease duration in years	3 (0–25)	4.5 (0–25)	0.8	3 (0–25)	3.5 (0–25)	0.54
Total serum IgE (kU/L)	55.8 (0–1186)	72.6 (2.0–415)	0.89	42.4 (1.98–233)	126 (12.4–1186)	**0.008**
Absolute basophil count (cells/µL)	5.8 (0.1–31.4)	7.1 (0.1–28.1)	0.99	3.2 (0.1–31.4)	12.4 (0.1–30.1)	**0.02**
Absolute lymphocyte count (cells/µL)	1528 (27–2835)	1418 (826–2088)	0.55	1443 (27–2835)	1458 (817–2424)	0.89
Absolute monocyte count (cells/µL)	192 (14–504)	194 (118–409)	0.54	192 (14–504)	194 (107–382)	0.41
Absolute granulocyte count (cells/µL)	3110 (152–6920)	2980 (1675–5752)	0.61	3571 (152–6920)	2872 (1675–4887)	0.15
BAT with donor basophils (% CD63+)	1.6 (0–94.0)	6.1 (0.8–11.1)	0.42	3.5 (0.0–94.0)	1.6 (0.3–59.4)	0.41

*Note:* Laboratory measurements and/or sampling were performed at baseline before the start of omalizumab treatment. Data are presented as median (range) unless otherwise described. Abbreviations: ai-S-MAT: autoimmune myeloma IgE sensitized MAT; ai-NS-MAT: autoimmune non-IgE sensitized MAT.

## Data Availability

The data supporting the findings of this study are available from the corresponding author upon reasonable request.

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
