# Peer review of "Autoimmune Mast Cell Activation Test as a Diagnostic Tool in Chronic Spontaneous Urticaria"

_ijms, 2024, doi:10.3390/ijms25179281_

Round 1
Reviewer 1 Report
Comments and Suggestions for Authors
ILMS 3149891
Comments:
In this paper, the authors endeavor to develop a novel autoimmune MAT utilizing the human mast cell line, LAD2. By employing LAD2 cells instead of donor-derived mast cells for MAT, there is a potential to ensure a more consistent quality of mast cells.
Although the number of patient samples analyzed seems somewhat limited, this study may contribute for advancing our understanding of the mechanisms that trigger CSU. An increase in sample size in future research would be highly desirable.
A few minor revisions are listed below.
1. Please explain the rationale for the two volumes (10 and 25 μL) of CSU patient /control serum added to the LAD2 cells.
2. In 2.2 of Results
1) Figure 1: The labels on the X-axis and Y-axis in the flow cytometry diagram are too small and unclear. Please enlarge them for improved clarity and include the scale on the X-axis.
2) In section 2.2, the numerical results of the analysis are provided, but the lack of a scale on the X-axis makes it difficult to understand. Please modify or supplement Figure 1 to include a scatter plot and the numerical values for each positive section.
Author Response
In this paper, the authors endeavor to develop a novel autoimmune MAT utilizing the human mast cell line, LAD2. By employing LAD2 cells instead of donor-derived mast cells for MAT, there is a potential to ensure a more consistent quality of mast cells.
Although the number of patient samples analyzed seems somewhat limited, this study may contribute for advancing our understanding of the mechanisms that trigger CSU. An increase in sample size in future research would be highly desirable.
Thank you very much for taking the time to review this manuscript. Yes, we agree that increasing the sample size would signify our findings. This would especially be beneficial in the case of omalizumab nonresponders, as we already mentioned in the discussion. We are planning to include more patients in our future studies. Please find the detailed responses below and the corresponding revisions/corrections highlighted/in track changes in the re-submitted files
A few minor revisions are listed below.
Comment 1: Please explain the rationale for the two volumes (10 and 25 μL) of CSU patient /control serum added to the LAD2 cells.
Response 1: In our initial experiments, performed in six patients, we tested different volumes (50 µl, 25 µl and 10 µl) of CSU patient plasma and found out that the 50 µl of serum was too high, therefore had an inhibitory effect on LAD2 cell CD63 activation (This data is not presented). We, therefore, decided to test the whole cohort for only 25 µl and 10 µl of CSU patient/control sera, which showed a dose-dependent response. We explained this information in the Methods section 4.3.
Comment 2: In 2.2 of Results
1) Figure 1: The labels on the X-axis and Y-axis in the flow cytometry diagram are too small and unclear. Please enlarge them for improved clarity and include the scale on the X-axis.
2) In section 2.2, the numerical results of the analysis are provided, but the lack of a scale on the X-axis makes it difficult to understand. Please modify or supplement Figure 1 to include a scatter plot and the numerical values for each positive section.
Response 2: 1) We enlarged the labels on the x-axis and y-axis and included the scale on both axes of Figure 1.
2) Thank you for this useful comment. Figure 1 shows an example of flow cytometric analysis of CD117, FcɛRI, and IgE expression on non-IgE-sensitized LAD2 cells and myeloma IgE-sensitized LAD2 cells. These measurements were performed for LAD2 quality control assessment in every batch of experiments (altogether four batches) and are not patient-specific. Since this result is not part of the diagnostic utility of autoimmune LAD2 MAT, we added additional subsection 2.2 LAD2 characteristics, and additional clarifications were made in the text.
Reviewer 2 Report
Comments and Suggestions for Authors
The introduction lacks a bibliographical reference in the phrase "Mast cell (MC) activation plays a significant role in CSU. Still, the exact cause of MC activation is unclear."; "These autoantibodies are thought to activate MCs and basophils via the IgE/FcɛRI pathway, releasing histamine and other inflammatory mediators, which cause the characteristic signs and symptoms." All sentences must have a bibliographical reference, verifying the same situation throughout the textual content.
"2.2. . The diagnostic utility of autoimmune LAD2 MAT" it should appear "2.2. The diagnostic utility of autoimmune LAD2 MAT".
In figure 1, the figure captions cannot be seen very well, they should be of better quality and larger captions. The figure does not present any statistical analysis (such as means, standard deviations, or p-values), which may be important for the reader to better analyze the results. Without this, it is difficult to assess the significance of the observed differences. Were the authors expecting differences between the graphs before and after IgE stimulation? It appears that the histograms of "IgE" and "IgE after stimulation" are very similar, which may suggest that the stimulation did not have the expected effect. If this is unexpected, it may indicate a problem with the experimental protocol or sample preparation.
In Figure 2, to remind the reader, in my opinion, somewhere in the figure the N used should appear.
Consideration should be given to adding error bars or confidence intervals to bar charts to highlight the variability and precision of the medians.
In the materials and methods, some numbers must be written in full. Normally, the rule is applied that up to 10, numbers are written in full, unless they appear at the beginning of the sentence. For example, "Except for two patients who received 3 subcutaneous doses of 150 mg omalizumab, all patients received 3 subcutaneous doses of 300 mg omalizumab at 4-week intervals." should be rephrased to "Except for two patients who received three subcutaneous doses of 150 mg omalizumab, all patients received three subcutaneous doses of 300 mg omalizumab at four-week intervals.". Whenever appropriate, the same rule should be applied to the remaining text.
Author Response
Comment 1: The introduction lacks a bibliographical reference in the phrase "Mast cell (MC) activation plays a significant role in CSU. Still, the exact cause of MC activation is unclear."; "These autoantibodies are thought to activate MCs and basophils via the IgE/FcɛRI pathway, releasing histamine and other inflammatory mediators, which cause the characteristic signs and symptoms." All sentences must have a bibliographical reference, verifying the same situation throughout the textual content.
Response 1: We added bibliographical reference for each sentence.
Comment 2: "2.2. . The diagnostic utility of autoimmune LAD2 MAT" it should appear "2.2. The diagnostic utility of autoimmune LAD2 MAT".
Response 2: We corrected this typing error.
Comment 3: In figure 1, the figure captions cannot be seen very well, they should be of better quality and larger captions. The figure does not present any statistical analysis (such as means, standard deviations, or p-values), which may be important for the reader to better analyze the results. Without this, it is difficult to assess the significance of the observed differences. Were the authors expecting differences between the graphs before and after IgE stimulation? It appears that the histograms of "IgE" and "IgE after stimulation" are very similar, which may suggest that the stimulation did not have the expected effect. If this is unexpected, it may indicate a problem with the experimental protocol or sample preparation.
Response 3: Thank you for this useful comment. We enlarged the labels on the x-axis and y-axis and included the scale on both axes of Figure 1. Figure 1 shows an example of flow cytometric analysis of CD117, FcɛRI, and IgE expression on non-IgE-sensitized LAD2 cells and myeloma IgE-sensitized LAD2 cells. These measurements were performed for LAD2 quality control assessment in every batch of experiments (altogether four batches) and are not patient-specific. Since this result is not part of the diagnostic utility of autoimmune LAD2 MAT, we added additional subsection 2.2 LAD2 characteristics, and additional clarifications were made in the text.
Regarding our expectations of differences between the histograms of “IgE” and “IgE after the stimulation”: When we designed this study, we predicted that NS-LAD2 would be a suitable model for the detection of anti-FcɛRI autoantibodies, while S-LAD2 would better detect anti-IgE autoantibodies. LAD2 quality control assessment confirmed that IgE from CSU patient/control sera did not yet bind to LAD2 cells during stimulation, which served as control of our experiment design. We provided additional clarifications and discussed these results in the Discussion.
Comment 4: In Figure 2, to remind the reader, in my opinion, somewhere in the figure the N used should appear.
Response 4: We added this information in Figure 2 and the Description of Figure 2.
Comment 5: Consideration should be given to adding error bars or confidence intervals to bar charts to highlight the variability and precision of the medians.
Response 5: The error bars (interquartile range) were changed to black for better visibility.
Comment 6: In the materials and methods, some numbers must be written in full. Normally, the rule is applied that up to 10, numbers are written in full, unless they appear at the beginning of the sentence. For example, "Except for two patients who received 3 subcutaneous doses of 150 mg omalizumab, all patients received 3 subcutaneous doses of 300 mg omalizumab at 4-week intervals." should be rephrased to "Except for two patients who received three subcutaneous doses of 150 mg omalizumab, all patients received three subcutaneous doses of 300 mg omalizumab at four-week intervals.". Whenever appropriate, the same rule should be applied to the remaining text.
Response 6: Thank you for this useful comment. We applied this rule where indicated.